# Diversity, Antimicrobial, Antioxidant, and Anticancer Activity of Culturable Fungal Endophyte Communities in *Cordia dichotoma*

**DOI:** 10.3390/molecules28196926

**Published:** 2023-10-04

**Authors:** Mahima Sharma, Sahil Bharti, Anindya Goswami, Sharada Mallubhotla

**Affiliations:** 1Tissue Culture Laboratory, School of Biotechnology, Shri Mata Vaishno Devi University, Kakryal, Katra 182320, India; 19dbt001@smvdu.ac.in; 2CSIR-Indian Institute of Integrative Medicine, Canal Road, Jammu 180001, India; sahilbharti.46@gmail.com (S.B.); agoswami@iiim.res.in (A.G.)

**Keywords:** endophytic organisms, bioactive compounds, anticancerous, antifungal, lasura, secondary metabolites

## Abstract

Endophytic fungi are a significant source of secondary metabolites, which are chemical compounds with biological activities. The present study emphasizes the first-time isolation and identification of such fungi and their pharmacological activities from the medicinal plant *Cordia dichotoma*, which is native to Jammu, India. The Shannon Wiener diversity index revealed a wide range of fungal endophytes in root (1.992), stem (1.645), and leaf (1.46) tissues. A total of 19 endophytic fungi belonging to nine different genera were isolated from this plant and the majority belonged to the Ascomycota phylum. ITS rRNA gene sequencing was used to identify the fungal strains and they were submitted in NCBI GenBank. The most potent fungal isolate *Cladosporium cladosporioides* OP870014 had strong antimicrobial, antioxidant, and anticancer activity against MCF-7, HCT-116, and PC-3 cancer cell lines. The LC-MS and GC-MS analyses of the ethyl acetate extract of *C. cladosporioides* were examined to identify the bioactive metabolites. The major compounds of the crude extract derived from *C. cladosporioides* OP870014, according to GC-MS, are spiculisporic acid; dibutyl phthalate; phenylethyl alcohol; cyclohexanone, 2,3,3-trimethyl-2-3-methylbutyl; pyrrolo[1,2-a]pyrazine-1,4-dione,hexahydro-3-(phenylmethyl);2,5-piperazinedione,3,6-bis(2-methylpropyl); and heneicosane which possessed antimicrobial, anticancerous, and antioxidant activities. The findings revealed that *C. dichotoma* has the capacity to host a wide variety of fungal endophytes and that secondary metabolites from the endophytic fungus may be a source of alternative naturally occurring antimicrobial, antioxidant, and cytotoxic compounds.

## 1. Introduction

The most crucial components of a plant’s microbiome are their endophytes. Although they live inside the plants in various relationships, these endophytes have no harmful effects on the plants [1]. Endophytes can be found inside the tissues of various plants all over the world and are widespread in nature. They reside within the plant and aid the plants in better adapting to their ecological niche [2]. Endophytes interact with each other and their host plant throughout this process, altering the plant’s capacity for metabolism [3]. Moreover, endophytes are well-known for producing numerous secondary metabolites with significant medicinal value [4]. Endophytic fungi can also produce various biologically active compounds that plants produce, i.e., taxol [5]. These metabolites have a wide range of bioactivities, i.e., insecticidal, antifungal, antidiabetic, antiviral, antioxidant, anticancer, antibacterial, etc., and belong to many structural classes [6]. Numerous natural medicines, including cyclosporine, cephalosporins, lovastatin, rapamycin, and paclitaxel, are still used to suppress the immune system and treat hypertension, lipid disorders, cancer, and parasite illnesses [7]. Research problems include the rise in cancer cases and the emergence of superbugs brought on by antibiotic resistance. Hence, it is crucial to find new and safer drugs based on natural products [8]. By choosing underutilized and unexplored natural resources, success can be attained [9]. They are anticipated to yield novel chemical entities. Among the most promising sources of novel and natural medications are endophytic fungi that live in a particular habitat [10].

The *Cordia dichotoma*, commonly named Lasura, is a member of the Boraginaceae family. It is a vastly distributed, medicinal tree that grows in tropical and subtropical regions of the world. It is indigenous to China, Pakistan, Thailand, Indonesia, Vietnam, and Australia. It is also widespread in several regions of India [11]. Its immature fruits are pickled, used as vegetables and fodder, and the leaves also yield good fodder. Its leaves are used to cure fever, headache, and joint pain, and its bark paste is useful to cure stomach ailments and reduce chest pain [12]. Generally, all of the plant’s parts have been employed to treat a variety of diseases including antihelmintic, antitumor, analgesic, antmicrobial, antifertility, wound healing, cough, dysentery, antiulcer, jaundice, and dyspepsia [13]. The economic significance and wide range of therapeutic applications of *C. dichotoma* have sparked interest in studying the endophytic fungi that have colonized this plant and their geographic range. Additionally, their anticancer properties and antimicrobial potential against a panel of clinically serious human pathogens were investigated. This endeavor is novel because there are not any known records of endophytic fungi from *C. dichotoma*, especially in the Jammu region.

## 2. Results and Discussion

### 2.1. Isolation, Diversity, Identification, and Characterization of Endophytic Fungi

In this study, a total of 169 fungal colonies (isolation rate, 36.7%) were isolated from 461 tissue segments of the *C. dichotoma* plant and included 85 (47.2%), 58 (34.5%), and 26 (23%) strains from root, stem, and leaf tissue samples, respectively. The 169 isolates were allocated to 14 distinct morphotypes [roots (8), stems (6)] based on their culture characteristics on SDA plates. A microscopical analysis and colony morphology were used to identify endophytes (Appendix A). Furthermore, the ITS-based rDNA sequence analysis was used to carry out their molecular identification. Table 1 provides information of endophytic fungi, including their GenBank accession numbers, isolation source, and nearest sequence homolog. The evolutionary relationships between these isolates and their related fungi are shown in Figure 1. The isolated fungal endophytes belonged to nine different genera and the majority belonged to the Ascomycota phylum. Two isolates were classified as Basidiomycota, represented by order Polyporales, comprising the two genera *Oligoporus* and *Rhizoctonia*. According to Shannon–Wiener diversity indices, *C. dichotoma* supports a variety of fungi, and the diversity of its endophytic communities is highest in the tissues of its roots (1.992), followed by their stems (1.645) and leaves (1.46) (Appendix A). In further analyses, morphotypes belonging to two classes of the Ascomycota phylum, including Dothideomycetes and Eurotiomycetes, were recognized. Most of the isolates belonged to the Dothideomycetes class in this study. Eurotiomycetes were represented by the order Eurotiales and three genera: *Penicillium*, *Talaromyces*, and *Aspergillus*; whereas Dothideomycetes comprised three orders: Pleosporales, Botryosphaeriales, and Capnodiales and four genera: *Alternaria*, *Lasiodiplodia*, *Epicoccum*, and *Cladosporium*.

The data presented here demonstrates that *C. dichotoma* roots possess a rich diversity of fungal endophytes and we found that Ascomycota, which is considered among the most common group of eukaryotes globally [14], was revealed to be the most common phylum of fungus. We also observed that the roots of *C. dichotoma* contain 42.10% of the endophytic fungi, whereas the stems and leaves only contain 31.57% and 26.31%, respectively. *Talaromyces* and *Alternaria* are common fungal genera and were abundant in the roots, stems, and leaves, whereas *Lasiodiplodia* and *Epicoccum* only colonized roots, while *Cladosporium* was only found associated with stems. The endophytes of *C. dichotoma* and *Lithospermum officinale* were compared because they both belong to the same family. Endophytes connected to *L. officinale* displayed the same class representation [15]. The endophytic communities of *C. dichotoma* and *L. officinale* may vary due to host specificity and their interactions with distinct ecological niches. The change in the spatial distribution of the endophytic community of the *C. dichotoma* indicates that diverse microenvironments of tissues play distinct roles in determining the composition of their microbiota. These findings are in line with earlier research on endophytic fungus isolated from numerous Indian medicinal plants [16].

### 2.2. Antimicrobial Activity

The antimicrobial potential of the selected fungal endophytes and plant parts (root, stem) were assessed against clinical strains of Gram-negative, Gram-positive bacteria, and pathogenic fungi by the agar disc-diffusion and tube dilution method. The extract of root, stem, and fungal endophytes which prohibited the growth of any test organisms was considered to have antimicrobial activity, and the zone of inhibition was determined in mm (Table 2). The results showed that endophytic fungi have significantly higher antimicrobial activity than the extract of the root and stem of *C. dichotoma*, even while both the plant root and stem extract did not show activity against *Salmonella typhi* and *Aspergillus niger*; however, some isolated endophytes showed contrary inhibition. Out of 14 endophytes screened, the majority showed antimicrobial activity against *E. coli* and *S. aureus*, followed by *B. subtilis*. Fungal strains, i.e., MCR2, MCR3, MCR8, MCS2, and MCS6, exhibited antimicrobial activity among all the nine test organisms (Appendix A). The best activity was exhibited by MCR3 against *S. aureus*, *E. coli*, *B. subtilis*, *C. tropicalis*, *S. typhi*, *A. niger*, *C. albicans*, *K. pneumonia*, and *P. aeruginosa* with the zone of the inhibition diameter of 34.8, 32.1, 30.5, 29, 28.2, 26.8, 24, 22, and 19.5 mm, respectively, followed by MCS2. 

The antimicrobial activity of the selected endophytes and plant parts (root, stem) was also determined by the tube dilution method (Table 3). Crude extracts of root, stem, and fungal isolates have shown MIC ranging from 200 to 3.125 µg/mL. In the present study, out of eight active extracts, the majority were found to be active against *B. subtilis* followed by *E. coli*. Extracts of MCR3 showed more potent MIC with at the least concentration of 6.25 µg/mL against *S. aureus*, whereas MCS2 showed antimicrobial potency MIC at the least concentration of 25 µg/mL against *B. subtilis*. However, numerous studies have demonstrated the ability of endophytic fungus from medicinal plants to combat microbes [17]. The MIC of endophytic fungi isolated from plants collected from Palolo, Central Sulawesi showed a MIC value of 256 µg/mL against *E. coli* and *S. aureus* [18]. The MIC of fungal endophytes isolated from the *Garcinia* plant species displayed the value of 512-2 µg/mL against *S. aureus*, *C albicans*, *C neoformans*, and *M gypseum* [19].

### 2.3. Total Phenolic and Flavonoid Content

Due to the aromatic rings on phenolic and flavonoid compounds and their derivatives, which lead to their antioxidant activities, these molecules have been regarded as the principal free radical scavengers [20]. There was a broad range of total phenol content (TPC) produced by the plant and fungal endophytes in their ethyl acetate (EA) extracts ranging from 12.3 to 138.4 µg GAE/mg of extract (Table 4). The highest phenolic content value was 138.4 µg GAE/mg extract for MCS2, followed by MCR6 (92.24 µg GAE/mg), whereas the least phenol content was noticed for MCR1 of 12.3 µg GAE/mg extract.

The result for total flavonoid content has shown a variation of approximately 13-fold, ranging from 7.8 µg catechin/mg to 105.4 µg catechin/mg of the EA extract of selected endophytic fungi and plant extract (Table 4). The highest flavonoid content has been shown by the EA extract of MCS2 with 105.4 µg catechin/mg of EA extract, whereas the lowest flavonoid content has been displayed by the plant root extract value of 7.8 µg catechin/mg of the EA extract. 

According to reports, phenolic substances shield biological systems from oxidative damage by oxygen scavenging, metal inactivation, free radical suppression, and peroxide breakdown [21]. The variation in phenolic and flavonoid patterns among various species of endophytic fungi directly influences their biological activities, including cytotoxic and antioxidant activities. It was found that the high flavonoid-producing fungal endophyte from *Conyza blini* has significant antibacterial and antioxidant activity [22]. Another report found that the phenolic (204, 312.3 µg GAE/mg of dry extract) and flavonoid (177.9 and 644.1 µg RE/mg of dry extract) content of the fungal isolates *A. alternata* and *C. cladosporioides* have a dose-dependent radical scavenging activity [23]. Our results further support the notion that the presence of phenol and flavonoid content corresponds with antioxidant and cytotoxic properties, pointing to *C. cladosporioides*’ EA extract as a potential treatment for a variety of disorders brought on by oxidative damage.

### 2.4. Antioxidant Activity

The antioxidant activity was detected visibly via color change from purple to yellow, while the inhibition percentage was measured through a spectrophotometric analysis at 517 nm. The antioxidant activity of the plant extract and fungal isolates were evaluated, and the results manifested that some of the strains had high antioxidant activity while the remaining strains had minimal antioxidant activity. Out of the nine fungal strains, MCS2 demonstrated best antioxidant activity as compared to the other strains and plant extracts (Table 4). The IC50 value for MCS2 was found to be 95.56 ± 0.4 µg/mL. Another strain, MCR6, displayed moderate antioxidant activity with an IC50 value of 131.24 ± 0.8 µg/mL, whereas MCR4 showed least antioxidant activity with an IC50 value of 341.97 ± 0.6 µg/mL. Ascorbic acid, a positive control, was used to compare the antioxidant potential of endophytic fungi-generated bioactive compounds. The IC50 value of ascorbic acid was calculated to be 18.24 ± 0.4 µg/mL using the DPPH assay.

Hydrogen peroxide is a mild oxidizer that can directly inhibit the activity of a few enzymes by oxidizing crucial thiol (-SH) groups. It can rapidly permeate cell membranes, and within the cell, it is likely to react with any molecule in a living organism, particularly DNA, lipids, and proteins, which in turn could be the source of many hazardous consequences [24]. The IC50 value of the plant extract and endophytic fungi is presented in (Table 4). The IC50 value of the standard ascorbic acid was (77.49 ± 0.4 µg/mL), while that of the fungal isolate MCS2 showed the highest antioxidant activity with an IC50 value of 149.51 ± 0.2 µg/mL, and the least antioxidant activity was exhibited by MCS4 with an IC50 value of 371.3 ± 1.2 µg/mL using the H_2_O_2_ assay.

Results showed that (MCS2) *C. cladosporioides*’ EA extract had a higher phenolic content, and it also had the best antioxidant activity compared to other strains. The antioxidant activity of the root and stem extract and H_2_O_2_ antioxidant activity of MCR1 did not correlate with its phenolic content, which suggested the presence of additional classes of compounds with antioxidant potential. Non-significant phenolic content produced by endophyte *A. brasiliensis* may be due to the assessment method for the substance or interference from other substances. Other aspects include the additive effects of several classes of substances with potential antioxidant properties [25,26]. The relationship between phenolic content and antioxidant activity has also been inconsistently correlated in some earlier investigations [27]. According to a prior study, the fungal endophyte *Cladosporium species*, isolated from five Sudanese medicinal plants, showed a linear correlation between a high phenol content and DPPH scavenging activity with an IC50 value of 1142.0 µg/mL [28]. Another report found that, *C. cladosporioides* isolated from *Taxus wallichiana* showed a linear correlation between high phenol content and DPPH scavenging activity with an IC50 value of 22.15 µg/mL [29]. These findings suggest that *C. cladosporioides*, a fungal endophyte isolated from *C. dichotoma*, may be a potential protector against oxidative damage.

### 2.5. Anticancerous Activity

Natural substances synthesized by fungal endophytes have recently shown to be a valuable source for drug discovery. Numerous endophytic fungi produce secondary metabolites that have anticancer properties [30]. The secondary metabolites produced by the fungal endophyte, *Peniophora incarnate*, exhibited in vitro cytotoxicity activity against three human cancer cells lines: HL-60, A375, and MCF-7 [31]. Another study reported that the endophytic fungus *Paramyrothecium roridum* isolated from *Morinda officinalis* showed cytotoxic activity against the human cancer cell lines epG-2, SF-268, and NCI-H460 [32]. In our current study, the crude extracts of the plant and nine endophytic fungi were screened for in vitro cytotoxicity activity against three human cancer cells lines: MCF-7 (Breast cancer), HCT-116 (Colon Cancer), and PC-3 (Prostate Cancer) (Table 5). All outcomes were compared with doxorubicin serving as a positive control. The crude extract of the endophytic fungus *Cladosporium cladosporioides* (MCS2) exhibited anticancerous activity against all the three cell lines and showed stronger activity than the positive control doxorubicin against one cancer cell line PC-3 with an IC50 value of 0.74 ± 0.008 µg/mL. The IC50 values of the extract of *Lasiodiplodia theobromae* (MCR3) displayed anticancerous activity against three cell lines: 1.19 ± 0.69 µg/mL against MCF-7, 1.045 ± 0.15 µg/mL against HCT-116, and 39.73 ± 1.19 µg/mL against PC-3. The crude extract of the endophytic fungus *Talaromyces purpureogenus* (MCS6) showed significant cytotoxic activity against HCT-116 with an IC50 value of 1.51 ± 0.16 µg/mL.

### 2.6. Chemical Composition of the Endophyic Fungus Cladosporium cladosporioides (MCS2) Analyzed by GC-MS

As mentioned above, we performed several biological assays using the crude extracts of isolated endophytic fungi. Out of all the extracts analyzed, *Cladosporium cladosporioides* (MCS2) showed remarkable pharmacological activity. We decided to study the chemical composition of the crude ethyl extract of *C. cladosporioides* (MCS2), which was analyzed using GC-MS. In total, 61 compounds were identified from the ethyl acetate extract of *C. cladosporioides* (MCS2) (Figure 2). Appendix A displays the molecular weight, molecular formula, and retention time. The compounds that employed a major percentage in the extract are spiculisporic acid (23.21%), dibutyl phthalate (11.46%), phenylethyl alcohol (8.67%), cyclohexanone, 2,3,3-trimethyl-2-3-methylbutyl (7.57%), pyrrolo[1,2-a]pyrazine-1,4-dione, hexahydro-3-(phenylmethyl (4.60%), 2,5-piperazinedione, 3,6-bis(2-methylpropyl) (3.56%), and heneicosane (3.21%) at various time intervals. The endophytic fungus *C. cladosporioides* OP870014 produced bioactive compounds with anticancer, antimicrobial, and antioxidant properties, according to GC-MS analyses. The majority of the compounds included in crude extracts of (MCS2) *C. cladosporioides* OP870014 are alkaloids, alcohols, hydrocarbons, terpenes, and their derivatives. These substances have been identified in endophytes that have been extracted from medicinal plants and are recognized for their therapeutic benefits [33]. Similarly, the endophytic fungus *Chaetomium globosum* produced volatile metabolites such as phenol, 2,4 bis(1,1dimethylethyl), 10-heneicosene, E-14-hexadecenal, and 3-eicosene and 1-heneicosanol and is a viable candidate for drug development [34]. Additionally, 6-pp(6 pentyl-2H pyrone-2-one), disulphide dimethyl, heneicosane, m-camphoren, thiopivalic acid, pthalic acid, and benzene derivatives have been described in *Aspergillus clavatonanicus* [35]. The ethyl acetate portion of the *Streptomyces* strain included higher concentrations of eicosane and dibutyl phthalate, both of which have antimicrobial properties [36].

### 2.7. Chemical Profiling of Cladosporium cladosporioides (MCS2) Fungal Metabolites by LCMS

LCMS is one of the modern and potent current techniques for identifying compounds present in biological samples [37]. In this study, the EA extract with good antimicrobial, antioxidant, and anticancer activity was evaluated by using the LC-MS technique. The retention time and mass spectra of the extract fractions were compared with those of authentic samples and mass spectra from the data library. Chromatograms of the ethyl acetate extract of *C. cladosporioides* (MCS2) are shown in Figure 3. The EA extract of MCS2 detected 17 bioactive compounds using the positive ionization mode. The major compounds with bio-active properties detected are (1) L-serine-O-sulfate, (2) Se-methyl-L-selenocysteine, (3) 3,5-dibromo-L-tyrosine, (4) quintozene, (5) doxorubicin, (6) estramustine, (7) disulfiram, (8) xylitol, (9) dodecanedioic acid, (10) 6-endo-hydroxycineole, (11) 10-formyldihydrofolate, (12) 5,10-methylenetetrahydrofolate, (13) prunasin, (14) linustatin, (15) delphinidin, (16) Se-adenosylselenomethionine, and (17) naringenin (Appendix A). Similarly, the endophytic fungus *Chaetomium ovaoascomatis* produced various metabolites such as grevilline B, austdiol, quinalizatrine, and questinol, and is a viable candidate for drug development [38].

## 3. Materials and Methods

### 3.1. Collection, Identification, and Authentication of Plant Sample

Matured, healthy *Cordia dichotoma* plants were arbitrarily selected from the Herbal Garden of Shri Mata Vaishno Devi University, Katra, Jammu and Kashmir (32.9418° N, 74.9541° E, elevation 754 m) between March–May 2020; then, they were transported to the lab and processed to lessen the contamination rate. The plant specimen was kindly identified by Dr. Harish Dutt (Assistant Professor) and Nitin Kumar Katoch (Curator of Herbarium), and the identified specimen has been deposited at the Herbaria of the Department of Botany, University of Jammu under accession Number 16613.

### 3.2. Isolation of Fungal Endophytes

Different tissues (root and stem) of the disease-free *C. dichotoma* plants were carefully excised by using a sterile scalpel. Firstly, these tissues were properly rinsed under running tap water, followed by rinsing with sterile distilled water. Surface sterilization procedures were performed on clean tissue pieces using 0.1% of Tween 20 (*v*/*v*), 4% sodium hypochlorite (*v*/*v*) for 10 min, and 70% ethanol (*v*/*v*) for 3 min. After each stage, they were finally washed three times in autoclave distilled water. After surface sterilization, tissues were dried on blotting sheets that had been cut into 6 mm diameter discs under laminar air flow and placed on Sabouraud dextrose agar (SDA) plates with 50 μg/mL of chloramphenicol to protect against bacterial contamination. In order to verify the efficacy of surface sterilization, 100 μL of the sterile distilled water that was used in the final wash of the sterilization procedure was then planted onto the SDA at the same time. Plates were monitored daily for fungal growth surrounding the plant samples after being incubated at 28 °C. On fresh SDA plates, the mycelia of the fungus that had begun to grow from the tissues were subcultured and preserved at 4 °C.

### 3.3. Characterization of Fungal Endophytes—Macroscopic and Microscopic-Based Identification

The selected fungal endophytes were identified based on the morphology noticed during their growth on SDA. They were stained with lactophenol blue in order to reveal the hyphae structure, conidia, conidiophores, and their pattern under a microscope. 

### 3.4. Molecular Based Identification

Fungal endophytic strains were finally identified using internal transcribed spacer (ITS)-based rDNA sequencing. Genomic DNA of endophytic fungal isolates was isolated using the protocol in which 50 mg of fungal mycelia was placed in liquid nitrogen and crushed to a fine powder. It was transferred to 1000 µL of the cetyltrimethylammonium bromide buffer, vortexed thoroughly, and incubated for 1 h at 60 °C [39]. Tubes were centrifuged at 10,000× *g* for 10 min followed by extraction with 700 µL of phenol, chloroform, and isoamyl alcohol (25:24:1). The upper aqueous layer was collected and DNA was precipitated with 800 µL of chilled isopropanol and the tubes were slowly inverted to mix the contents followed by centrifugation at 10,000× *g* for 10 min at 4 °C. Thereafter, pellets were washed with chilled 70% ethanol followed by air drying. The dried pellets were eluted in an 80 µL TE buffer. A NanoDrop spectrophotometer was used to detect the absorbance at 260 nm to estimate the concentration of genomic DNA. The primers ITS1 (5′-TCCGTAGGTGAACCTGCGG-3′) and ITS4 (5′TCCTCCGCTTATTGATATGC-3′) were used to amplify 500–600 bp of the fungal genomic DNA [40]. The PCR reaction was set up in a total volume of 40 μL containing DNA (50–500 ng), which included 20 μL of the DreamTaq PCR master mix, 3 μL of template DNA, 2 μL of 10 nm both forward and reverse primer, and 13 μL of 10× buffer. Cycling parameters were one cycle of 95 °C for 5 min, followed by 32 cycles of 94 °C for 30 s, 58 °C for 1 min, 72 °C for 1 min, and the final extension of 15 min at 72 °C. The PCR product was separated using agarose gel electrophoresis at 95 V and visualized under UV light. The amplified product was purified using the Genei PureTM quick PCR purification kit and measured at 260 nm using a spectrophotometer. The purified partial ITS-based rDNA amplicons were sequenced by Biologia Research India. To establish sequence homology with closely related taxa, the sequences were assembled, modified, and aligned in MEGA11 before being compared to the GenBank database using the Basic Local Alignment Search Tool (https://blast.ncbi.nlm.nih.gov/Blast.cgi, accessed on 11 January 2022). The microorganisms with maximum similarity (100%) were selected as the closest match in this investigation, and isolated fungi were identified to the species level using the information from the closest related microbes.

### 3.5. Bioactivity Assessment Fermentation and Extraction

For the extraction of molecules, the isolated fungal endophytes were cultured in SDB in 250 mL flasks having 100 mL of the medium and incubated for 15 days at 28 °C under continuous shaking at 150 rpm. After incubation, broth carrying fungal growth was filtered with a muslin cloth. The CFS was obtained by centrifugation for 10 min at 12,000 rpm. Equal volumes of ethyl acetate (EA) were added to the filtrate, vigorously mixed, and left for 15 min until two clear immiscible layers were formed and by using a separating funnel, the upper EA layer was separated. Plant tissues were rinsed with running tap water, air-dried, coarsely powdered, and extracted for secondary metabolites with ethyl acetate using a soxhlet apparatus. The plant and fungal extracts were concentrated by using a rotary vacuum evaporator at room temperature, to obtain crude extracts which were dissolved in dimethyl sulphoxide (DMSO) at a concentration of 1 mg/mL and stored at 4 °C for subsequent research.

### 3.6. Estimation of Total Phenolic and Total Flavonoid Content

The Folin-Ciocalteu (FC) colorimetric technique was used to calculate total phenolic content. Plant extracts (100 μL) and endophytes were mixed with 550 μL of FC reagent and 500 μL of sodium carbonate solution (20%). After incubation for 30 min at room temperature in the dark, the absorbance was measured with a spectrophotometer at 650 nm. A calibration curve was created using a linear fit with gallic acid concentrations ranging from 10 to 100 µg/mL. The total phenol content was determined using the gallic acid calibration curve (y = 0.011x + 0.004), R^2^ = 0.992, where y represents absorbance and x represents concentration (µg/mg). The total phenol content was revealed as µg of gallic acid/mg of extract and the results were shown as the mean ± SD. 

The total flavonoid content was estimated using the aluminium chloride colorimetric method. The crude extract (1 mL) was mixed with an AlCl_3_ solution and placed in the dark for 10 min. The absorbance was taken at 430 nm, and the total flavonoid content was determined using the catechin calibration curve (y = 0.010x + 0.044), R^2^ = 0.997, where y represents absorbance and x is concentration (µg/mg). The experiment was performed in triplicate and the findings were showed as µg catechin/mg of extract, with the mean ± SD.

### 3.7. Antimicrobial Activity Agar Disc Diffusion Method

The antimicrobial activity of isolated endophytic fungi and the extracts of plant parts (root and stem) were tested against pathogenic bacteria and fungal strains, which included *Staphylococcus aureus* MTCC 737, *Pseudomonas aeruginosa* MTCC 1688, *Klebsiella pneumonia* MTCC 432, *Escherichia coli* MTCC 1687, *Bacillus subtilis* MTCC 1789, *Salmonella typhi MTCC 733*, *Aspergillus niger* MTCC 514, *Candida albicans* MTCC 854, and *Candida tropicalis* MTCC 461 using the agar disc diffusion method. The test organism cultures were placed across the surface of the Muller Hinton agar (MHA) plates using sterile cotton swabs. Ethyl acetate (80 µL), and the crude extract (1 mg/mL) of plant parts-root, stem, and fungal endophytes were impregnated onto sterile discs and placed on MHA plates. The plates were incubated at 37 °C for 24 h (bacteria) and at 28 °C for 48 h (fungi) and the inhibition zone was measured. The experiment was performed in triplicates. Chloramphenicol (30 μg/disc) and fluconazole (30 μg/disc) were taken as a positive control for bacterial and fungal strains, respectively, while DMSO was used as a negative control.

### 3.8. Tube Dilution Assay

The minimum inhibitory concentration (MIC) of fungal endophytes and the extracts of plant parts (root and stem) were estimated by the serial dilution method. MHB and SDB were made and sterilized using an autoclave. Prepared broth (1 mL) was placed into the tubes (marked 1 to 5) using a sterile syringe needle. A stock solution carrying 200 µg/mL of crude extracts of both plant parts (root, stem) and fungal isolates was prepared. The sterile MHB and SDB with 200 µg/mL of crude extract was diluted 2-fold seven times in sterile tubes aseptically. Tube 6 was taken as a control to check the sterility of the media and tube 7 for the viability of the test organisms. Subsequently, each tube was inoculated with an equal volume of the overnight culture, and the test organisms were prepared in sterile MHB and SDB. The final concentrations of the extract in each of the test tubes after dilution were 200, 100, 50, 25, 12.5, 6.25, and 3.125 µg/mL, and they were placed at 37 °C (bacteria) and 28 °C (fungi) for 24 h and were observed for growth. The lowest dilution tube in which growth did not occur was the MIC of the culture.

### 3.9. Antioxidant Activity DPPH Free Radical Scavenging Assay

The antioxidant activity of the plant extract and endophytic fungi were evaluated by the 2,2-diphenyl-1-picrylhydrazyl (DPPH) method and compared with well-known antioxidant ascorbic acid. The extract of plant and endophytic fungi was diluted to obtain concentrations of 25, 50, 75, 100, 150, and 200 μg/mL. An amount of 100 mL of 0.004% of the DPPH solution was prepared in 95% methanol. An amount of 1 mL of each diluted sample was mixed with 2 mL of the DPPH solution and incubated for 1 h in dark at room temperature. After incubation, absorbance was measured at 517 nm. An amount of 2 mL of 95% methanol mixed with 1 mL of methanol was used as a blank. An amount of 1 mL methanol and 2 mL of the DPPH solution were taken as the control. The free radical inhibiting % and IC50 value was calculated:(1)$$Freer\;adical\;scavenging=\frac{Acontrol-Asample}{Acontrol}\times 100.$$

### 3.10. Hydrogen Peroxide Scavenging Assay

The antioxidant activity of plant and endophytic fungi were evaluated by a H_2_O_2_ scavenging assay. A solution of H_2_O_2_ (40 mM) was made in a phosphate buffer. The crude extract (1 mL) of plant and fungi at a concentration of 25–200 μg/mL was added to equal the volume of the H_2_O_2_ solution and the total volume was made up to 3 mL. The absorbance was measured at 230 nm in a spectrophotometer. A blank containing a phosphate buffer, without H_2_O_2_, was prepared. The extent of H_2_O_2_ scavenging of the extracts of endophytic fungi was calculated as:H_2_O_2_ scavenging activity (%) = (A_blank_ − A_sample_)/A_blank_ × 100.(2)

### 3.11. Cytotoxic Activity

The MTT assay was used to test the cytotoxic activity of the extracts of plant and endophytic fungi. A colorimetric assay is used to measure the rate of cell survival and growth. Three human cancer cell lines: MCF-7 (Breast), HCT-116 (Colon), and PC-3 (Prostate) were procured from the American Type Culture Collection (Manassas, VA, USA) for the present study. Cells were cultured in RPMI-1640 media supplemented with 10% fetal bovine serum (FBS) and 100U penicillin G/100 μg mL^−1^ streptomycin. Cells were incubated at 37 °C with 98% humidity and 5% CO_2_. Cells (5 × 10^3^ cells per well) were seeded into a 96-well plate with suitable media and left to adhere overnight. The media was changed the following morning, and 200 µL of serum containing fresh media was added to each well. Plant and fungal extracts were serially diluted (500, 50, 5 and 0.5 ug/mL) in triplicate wells until the final concentration of the DMSO solvent was 0.5%. As a negative control, DMSO was used, and Doxorubicin Chloride was used as a positive control. After 48 h of incubation, 3-(4,5-dimethylthiazol-2-yl)-2,5-diphenyl tetrazolium bromide (MTT, 5 mg/mL) was added to each well and the cells were incubated at 37 °C for 4 h. The generated formazan crystals were dissolved by adding DMSO. After 15 min of incubation at room temperature, the amount of colored formazan derivatives was evaluated by measuring optical density (OD) at 570 nm with a microplate reader. GraphPad Prism software was used to calculate the IC50 values and ascertain the percentage of cell viability.

### 3.12. Chemical Constituents Using an GC-MS Analysis

The EA extract of the fungal isolate *Cladosporium cladosporioides* (MCS2), displaying significant antimicrobial, antioxidant, and anticancerous activity, was subjected to Gas Chromatography-Mass Spectrometry (GC-MS), in order to identify the many volatile bioactive compounds. CytoGene Research & Development performed the GC-MS analysis. SHIMADZU, the QP2010 model, was used for the GC-MS analysis. The sample injection was 2 μL, the injecting temperature was 280 °C, the pressure was 29.7 kPa, and the column flow rate was 0.72 mL/min. The running time was 52 min. The NIST14.L library (2020) was searched to compare the structures of compounds with that of the NIST database. Compounds were identified based on mass spectra and retention times with already familiar compounds in the NIST library (C:\Database\NIST14.L).

### 3.13. Chemical Profiling of C. cladosporioides (MCS2) Metabolites by LC-MS Analysis

Based on the antimicrobial, antioxidant, and anticancerous activities of the *C. cladosporioides* ethyl acetate extract, the sample was subjected to LC-MS (Waters, Xevo TQD, Milford, MA, USA). An exterior electrode and a core electrode, which serve as the analyzer and detector, make up the analytical device. During the study, the data processing software used was MzMine 2.53. Direct infusion was done using the positive ionization mode with a mass (*m*/*z*) range of 50 to 8000 amu. The column material C18 waters, Acquity BEH 2.1 × 100 mm, 1.7 µm, was used in conjunction with a mobile phase of 0.1% formic acid in the water solvent and acetonitrile solvent. The sample injection was 5 μL, the desolvation temperature was 550 °C, the source temperature was 120 °C, the pressure was 6–7 bar, and the column flow rate was 0.2 mL/min. The LCMS analysis was done at CytoGene Research & Development, Institute in Lucknow, Uttar Pradesh.

### 3.14. Statistical Analysis

The total number of plant samples colonized by fungi divided by the total number of samples incubated was used to calculate the colonization rate. The ratio of the number of fungal strains isolated from plant samples to the total number of segments incubated was used to compute the isolation rate. The following formula was used to construct the Shannon Weiner diversity index:*H*/ = −∑ [(pi) × log(pi)](3)
where pi represents the relative abundance of species and i indicates the total number of species [13]. The data shown are averages of the results from each experiment, which were all run in triplicate. The means and standard deviation were calculated using the SPSS-22 statistical program (SPSS, Inc., Chicago, IL, USA).

## 4. Conclusions

This is the first report of diversity of endophytic fungi isolated from *C. dichotoma*, specifically, in Jammu and Kashmir and our results indicate the high diversity of endophytic fungi associated with the root, stem, and leaf of *C. dichotoma* that exhibited anticancer, antioxidant, and antimicrobial activities. Among the various fungal isolates, extracts of *Cladosporium cladosporioides* (MCS2) were found to have strong cytotoxic, antioxidant, and antimicrobial properties which can thus serve as a source of novel cytotoxic/antioxidant/antimicrobial compounds. Bioactive secondary metabolites, which were produced by the endophytic fungi, *C. cladosporioides* (MCS2), were studied and identified through GC-MS and LC-MS analyses. The compounds produced by the most active fungal isolate *Cladosporium cladosporioides* (MCS2) exhibited cytotoxicity against MCF-7, HCT-116, and PC-3 cancer cell lines, in addition to displaying antimicrobial and antioxidant activities, indicating that it should be taken into consideration as a potential drug candidate with therapeutic properties.

## Figures and Tables

**Figure 1 molecules-28-06926-f001:**
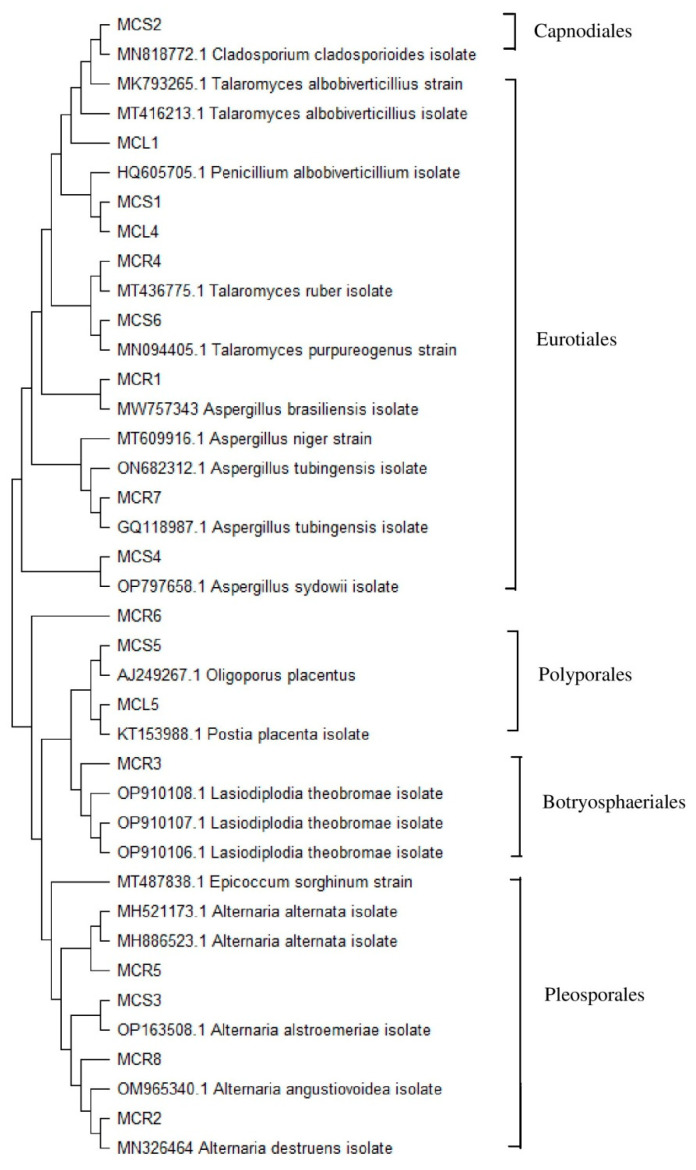
Phylogenetic tree from analysis of ITS rRNA gene sequences of the endophytic fungal strains associated with *Cordia dichotoma* using the maximum likelihood approach, the bootstrap consensus tree inferred from 1000 iterations, and a phylogenetic tree constructed using MEGA 11 software.

**Figure 2 molecules-28-06926-f002:**
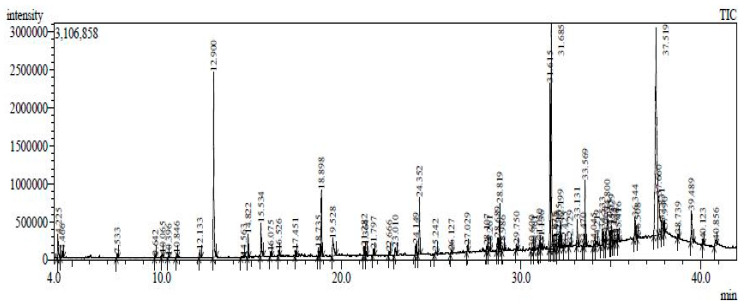
Abundance of the chemical compounds present in the ethyl acetate extract of *Cladosporium cladosporioides* (MCS2).

**Figure 3 molecules-28-06926-f003:**
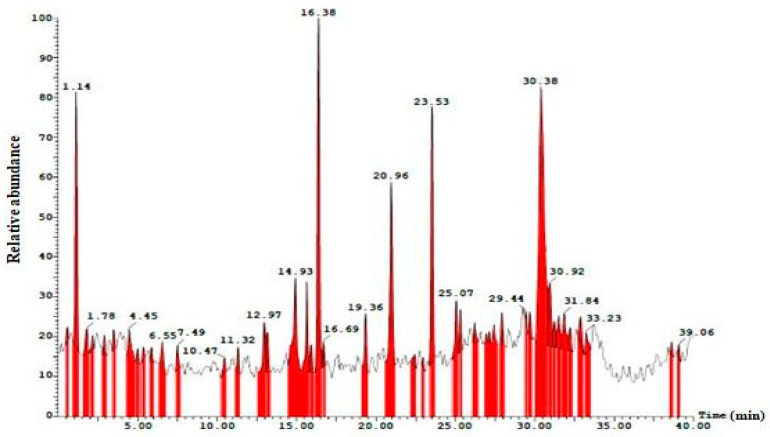
LCMS chromatogram of metabolites of *Cladosporium cladosporioides* (MCS2).

**Table 1 molecules-28-06926-t001:** Isolate code, GenBank accession number, closest related species, maximum identities, and identification of the endophytic fungi associated with *C. dichotoma*.

Tissue	Fungal Codes	GenBank Accession Number	Closest Relatives in NCBI	Reference Sequence Accession Number	Identity (%)
Root	MCR1	OP868830	*Aspergillus brasiliensis*	MW757343	94
	MCR2	OP868835	*Alternaria destruens*	MN326464	100
	MCR3	OP868836	*Lasiodiplodiatheobromae*	OP910107	100
	MCR4	OP868948	*Talaromycesruber*	MT436775	100
	MCR5	OP868964	*Alternaria alternate*	MH521173	100
	MCR6	OP869857	*Epicoccumsorghinum*	MT487838	100
	MCR7	OP870013	*Aspergillus tubingensis*	ON682312	99
	MCR8	OQ028266	*Alternaria angustiovoidea*	OM965340	99
Stem	MCS1	OP870014	*Aspergillus niger*	MT609916	100
	MCS2	OP870034	*Cladosporium cladosporioides*	MN818772	100
	MCS3	OP999368	*Alternaria alstroemeriae*	OP163508	99
	MCS4	OP870142	*Aspergillus sydowii*	OP797658	100
	MCS5	OP876791	*Oligoporus placenta*	AJ249267	97
	MCS6	OP870080	*Talaromycespurpureogenus*	MN094405	99

**Table 2 molecules-28-06926-t002:** Antimicrobial activity of endophytic fungi from *C. dichotoma* and the extract of plant parts, root and stem, against nine pathogens.

Fungal Isolate	Zone of Inhibition in mm								
	Test Organisms								
	*Bacillus subtilis*	*Escherichia coli*	*Staphylococcus* *aureus*	*Klebsiella* *pneumoniae*	*Salmonella* *typhi*	*Pseudomonas* *aeruginosa*	*Aspergillus* *niger*	*Candida* *albicans*	*Candida tropicalis*
MCR1	15.5 ± 0.4	13.5 ± 0.4	14 ± 0.8	9 ± 0.8	13.5 ± 0.4	14 ± 0.8	-	18 ± 0.4	19.8 ± 0.2
MCR2	22.1 ± 0.5	26.5 ± 0.4	19.9 ± 0.3	11.9 ± 0.3	16 ± 0.4	17.5 ± 0.4	23.8 ± 0.6	28 ± 0.8	16 ± 0.8
MCR3	30.5 ± 0.4	32.1 ± 0.2	34.8 ± 0.2	22 ± 0.5	28.2 ± 0.1	19.5 ± 0.4	26.8 ± 0.2	24 ± 0.2	29 ± 0.2
MCR4	10.8 ± 0.6	16 ± 0.4	16.5 ± 0.4	-	14.5 ± 0.4	-	14 ± 0.4	-	-
MCR5	-	13.8 ± 0.6	11.5 ± 0.4	9 ± 0.4	-	6.1 ± 0.2	-	11 ± 0.4	-
MCR6	23.8 ± 0.2	21.9 ± 0.3	18.5 ± 0.4	15 ± 0.4	-	17 ± 0.4	17.5 ± 0.4	-	19.9 ± 0.3
MCR7	13 ± 0.4	18.1 ± 0.1	20.7 ± 0.2	-	15 ± 0.8	-	-	14 ± 0.8	11.8 ± 0.5
MCR8	19.1 ± 0.6	22.5 ± 0.4	13.5 ± 0.4	17.6 ± 0.5	14.8 ± 0.6	16.1 ± 0.2	20.8 ± 0.6	17 ± 0.8	16.1 ± 0.2
MCS1	8.6 ± 0.6	10.5 ± 0.4	15 ± 0.4	12.5 ± 0.4	11.5 ± 0.4	-	-	8.1 ± 0.2	11 ± 0.4
MCS2	24 ± 0.8	28 ± 0.8	24 ± 1.6	25.5 ± 0.6	27.3 ± 0.8	21 ± 0.8	26.5 ± 2.0	18.4 ± 1.6	22 ± 0.8
MCS3	-	10 ± 0.4	-	14.5 ± 0.4	-	11.5 ± 0.4	-	-	-
MCS4	10.8 ± 0.6	11.5 ± 0.4	15.8 ± 0.6	-	-	12 ± 0.8	-	9 ± 0.4	-
MCS5	18.6 ± 0.6	-	17.5 ± 0.4	12 ± 0.8	-	-	-	13 ± 0.4	11.1 ± 0.2
MCS6	22.5 ± 0.4	24.8 ± 0.6	26.6 ± 1.24	19 ± 0.4	20.5 ± 0.6	15.1 ± 0.6	17 ± 0.8	21 ± 0.8	24.8 ± 0.6
Root	11.4 ± 0.9	14.7 ± 0.5	4.6 ± 1.6	3.2 ± 0.8	-	-	-	7.9 ± 1.3	12.3 ± 0.4
Stem	9.3 ± 0.7	5.8 ± 1.4	-	-	-	2.8 ± 0.6	-	10.9 ± 0.4	-
Chloramphenicol	31.5 ± 0.4	32.6 ± 0.4	35 ± 0.2	27.3 ± 0.4	28.6 ± 0.4	24.8 ± 0.6			
Fluconazol.e							29.2 ± 0.2	29.5 ± 0.4	30 ± 0.4
DMSO	-	-	-	-	-	-	-	-	-

**Table 3 molecules-28-06926-t003:** Minimum inhibitory concentration of fungal endophytes from *C. dichotoma* and the extract of plant parts, root and stem.

Fungal Isolate	Minimum Inhibitory Concentration (µg/mL)								
	Test Organisms								
	*Bacillus subtilis*	*Escherichia coli*	*Staphylococcus* *aureus*	*Klebsiella* *pneumoniae*	*Salmonella* *typhi*	*Pseudomonas* *aeruginosa*	*Aspergillus* *niger*	*Candida* *albicans*	*Candida tropicalis*
MCR1	-	-	-	-	-	-	-	-	-
MCR2	100	-	-	-	-	-	-	100	-
MCR3	25	12.5	6.25	200	50	100	-	50	100
MCR4	-	-	-	-	-	-	-	-	-
MCR5	-	-	-	-	-	-	-	-	-
MCR6	100	-	-	-	-	-	-	-	-
MCR7	-	-	200	-	100	-	-	-	-
MCR8	-	200	-	-	-	-	-	200	100
MCS1	-	-	-	-	-	-	-	-	-
MCS2	25	100	-	100	50	-	100	-	200
MCS3	-	-	-	-	-	-	-	-	-
MCS4	-	-	-	-	-	-	-	-	-
MCS5	-	-	200	-	-	-	-	-	-
MCS6	100	200	-	-	100	-	-	200	-
Root	-	200	-	-	-	-	-	-	-
Stem	-	-	-	-	-	-	-	-	-
Chloramphenicol	6.25	6.25	3.125	25	12.5	25			
Fluconazole							6.25	25	25

**Table 4 molecules-28-06926-t004:** Total phenolic content, flavonoid content, and antioxidant activity of plant extract and fungal strains associated with the *C. dichotoma*.

Fungal Isolate	Species Identified	Total Phenolic Content (µg GAE/mg Extract)	Total Flavonoid Content (µg Catechin/mg of Extract)	Antioxidant Activity IC50 Value (µg/mL)	
				DPPH Free Radical Scavenging Assay	H_2_O_2_ Scavenging Activity
MCR1	*Aspergillus brasiliensis*	12.3 ± 1.2	17.5 ± 2.0	321.21 ± 0.8	252.48 ± 2.6
MCR2	*Alternaria destruens*	50.18 ± 1.65	38.1 ± 0.8	198.09 ± 1.2	200.43 ± 0.5
MCR3	*Lasiodiplodiatheobromae*	56.4 ± 3.3	35 ± 2.1	175.82 ± 0.2	165.56 ± 0.2
MCR4	*Talaromycesruber*	22.1 ± 2.9	16.9 ± 2.4	341.97 ± 0.6	322.5 ± 0.9
MCR6	*Epicoccumsorghinum*	92.24 ± 2.9	64.1 ± 3.1	131.24 ± 0.8	156.63 ± 0.4
MCS1	*Aspergillus niger*	24.9 ± 2.1	23.4 ± 2.4	256.81 ± 0.4	269.02 ± 0.8
MCS2	*Cladosporium cladosporioides*	138.4 ± 1.6	105.4 ± 2.3	95.56 ± 0.4	149.51 ± 0.2
MCS4	*Aspergillus sydowii*	18.2 ± 2.8	33.8 ± 3.6	313.64 ± 0.2	371.3 ± 1.2
MCS6	*Talaromycespurpureogenus*	30 ± 2.4	18.7 ± 2.0	232.09 ± 1.5	301.79 ± 3.1
	Root	14 ± 0.6	7.8 ± 1.6	248.20 ± 0.8	302.59 ± 0.4
	Stem	23.5 ± 1.2	18.9 ± 0.8	224.92 ± 1.2	264.47 ± 1.6
	Positive control (Ascorbic acid)			18.24 ± 0.4	77.49 ± 0.2

**Table 5 molecules-28-06926-t005:** Cytotoxic activity of plant extract and fungal endophytes from *C. dichotoma* against human cancer cell lines MCF-7, HCT-116, and PC-3.

Fungal Isolate	Species Identified		IC50 (µg/mL) ± SD	
		MCF-7	HCT-116	PC-3
MCR1	*Aspergillus brasiliensis*	25.76 ± 1.23	>100	>100
MCR2	*Alternaria destruens*	5.85 ± 0.99	3.8 ± 0.46	>100
MCR3	*Lasiodiplodia theobromae*	1.19 ± 0.69	1.045 ± 0.15	39.73 ± 1.19
MCR4	*Talaromyces ruber*	>100	>100	>100
MCR6	*Epicoccum sorghinum*	2.82 ± 0.63	19.70 ± 1.22	>100
MCS1	*Aspergillus niger*	15.01 ± 1.99	15.81 ± 1.11	>100
MCS2	*Cladosporium cladosporioides*	3.96 ± 0.13	2.29 ± 0.16	0.74 ± 0.008
MCS4	*Aspergillus sydowii*	11.58 ± 1.22	>100	>100
MCS6	*Talaromyces purpureogenus*	7.37 ± 0.66	1.51 ± 0.16	>100
	Doxorubicin (Positive Control)	0.11 ± 0.01	0.6 ± 0.02	1.02 ± 0.05
	Root	23.4 ± 1.12	81.45 ± 1.22	>100
	Stem	>100	>100	>100

## Data Availability

Data is contained within the article or Appendix A.

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
