# Peer review of "Diversity, Antimicrobial, Antioxidant, and Anticancer Activity of Culturable Fungal Endophyte Communities in Cordia dichotoma"

_molecules, 2023, doi:10.3390/molecules28196926_

Round 1

Reviewer 1 Report

From the study of the manuscript it follows that the species Cordia dichotoma owes all its pharmaceutical qualities to the endophytic communities. Extracts made directly from the plant have not been analyzed? It would have been a very suitable control for all analyzes performed on therapeutic properties.

The molecular identification part lacks bibliographic references and correlation with morphological identification.

Table 1 is irrelevant to the paper since the data presented are described in the text. It can be used as additional material, but the abbreviations should be explained.

the identification of endophytic fungal species is relevant but I did not understand what was the purpose of generating the phylogenetic tree?

The results of the GC-MS and LC-MS analyzes shown in Tables 7 and 8 are far too explicit for a scientific article - the compound name, chemical formula and molecular mass is already too much. It is far too general information and irrelevant to the research being conducted.

I identified some English editing errors (e.g.. L 32 - they are endophytes or L 54).

Author Response

We are grateful to reviewer for the encouraging comments as we believe incorporation of these changes will enhance the quality of our manuscript. The changes suggested by reviewer 1 are highlighted in yellow color.

  1. As advised by the Reviewer, the data related to pharmaceutical activity of root and stem extract of plant has been incorporated in the revised manuscript.
  2. As advised by the Reviewer, the changes have been included in the revised manuscript.
  3. As suggested by the Reviewer, Table 1 has been deleted from the main file and added in the supplementary file.
  4. The purpose of generating the phylogenetic tree, is to understand the evolutionary relationship between endophytic fungi isolated in this study.
  5. As suggested by the Reviewer, the results of the GC-MS and LC-MS analyzes have been transferred to the supplementary file and removed from the main file. We think that sometimes, this data is useful for researchers.
  6. We are grateful for the remarks of the Reviewer, the suggested changes have been incorporated in the revised manuscript.

Reviewer 2 Report

1. Ê»It is indigenous to China, Taiwan, Pakistan, Thailand, Indonesia, Vietnam, and Australia.’ In this sentence, I need to emphasize that since ancient times, Taiwan has been an inseparable part of China, and Chinese sovereignty and territorial integrity cannot be divided. Therefore, Taiwan belongs to China and cannot be a parallel relationship. This is a very important point.

2. There is no experimental procedure for isolating fungal colonies from Cordia dichotoma plant, and only the results of 19 different morphological colonies were obtained.

3. You wrote in your paper that five fungal colonies were extracted from the leaves of Cordia dichotoma, but there were only three in your attached picture, and these two fungi did not appear in the following experiments. If you have the pictures and data of these two fungi, please put them up, if not, please delete them.

4. When you write fungi in the text, you use the name in the column of Closest relatives in NCBI. It is suggested to change it into fungal code, which is more convenient for readers to read and deepen their influence.

5. Ê»and least antioxidant activity showed by Aspergillus sydowii with IC50 value of 301.79±3.1 µg/mL using H2O2 assay .ʼ According to your table, the Aspergillus sydowii data of MCS4 is 371.3±1.2, please correct the error here.

6. Your essay is well written and has a lot of content, but it could be more concise and better.

Ê»It is indigenous to China, Taiwan, Pakistan, Thailand, Indonesia, Vietnam, and Australia.’ In this sentence, I need to emphasize that since ancient times, Taiwan has been an inseparable part of China, and Chinese sovereignty and territorial integrity cannot be divided. Therefore, Taiwan belongs to China and cannot be a parallel relationship. This is a very important point.

Author Response

We are grateful to reviewer for the encouraging comments as we believe incorporation of these changes will enhance the quality of our manuscript. The changes suggested by reviewer 2 are highlighted in green color.

  1. Thanks to the Reviewer for the advice. We apologize for this mistake and changes have been incorporated in the revised manuscript.
  2. The experimental procedure for isolation of fungal colonies from Cordia dichotoma plant are presented in the material and method section under the Subheading: Isolation of fungal endophytes.
  3. As suggested by the Reviewer, the details have been included in the revised manuscript.
  4. As suggested by the Reviewer, the changes have been incorporated in the revised manuscript accordingly.
  5. As suggested by the Reviewer, the error has been corrected in the revised manuscript.
  6. As suggested by the Reviewer, the manuscript has been made more concise and better in the revised manuscript.

Round 2

Reviewer 1 Report

I declare myself satisfied with the changes and congratulate you!

I also identified a few small problems with expression and editing. I hope that these will be corrected in the proof reading stage.

Author Response

We are grateful to Reviewer for the encouraging comments as we believe incorporation of these changes will enhance the quality of our manuscript.

As advised by the Reviewer, the changes have been incorporated in the revised manuscript as were found appropriate.

Reviewer 2 Report

Ê»It is indigenous to East Asian countries, Pakistan, Thailand, Indonesia, Vietnam, and Australia.ʼ In this sentence, the East Asian countries include China, Japan, South Korea, North Korea and Inner Mongolia, which is not consistent with the origin of China that you first wrote. I think it is too vague and the origin is not accurate.

Author Response

We are grateful to Reviewer for the very relevant comment. The changes suggested by Reviewer 2 have been accordingly incorporated in the manuscript and are highlighted in grey color.

Thank you for your suggestion; we replaced the phrase “East Asian countries” with China in the revised manuscript, in order to make the origin of the plant species in the region clear.  
